# Unlearning in- vs. out-of-distribution data in LLMs under gradient-based methods

**Teodora Baluta** *
teobaluta@gatech.edu

**Pascal Lamblin** †
lamblinp@google.com

**Daniel Tarlow** †
dtarlow@google.com

**Fabian Pedregosa** †
pedregosa@google.com

**Gintare Karolina Dziugaite** †
gkd@google.com

## Abstract

Machine unlearning aims to solve the problem of removing the influence of selected training examples from a learned model. Despite the increasing attention to this problem, it remains an open research question how to evaluate unlearning in large language models (LLMs), and what are the critical properties of the data to be unlearned that affect the quality and efficiency of unlearning. This work formalizes a metric to evaluate unlearning quality in generative models, and uses it to assess the trade-offs between unlearning quality and performance. We demonstrate that unlearning out-of-distribution examples requires more unlearning steps but overall presents a better trade-off overall. For in-distribution examples, however, we observe a rapid decay in performance as unlearning progresses. We further evaluate how example's memorization and difficulty affect unlearning under a classical gradient ascent-based approach.

## 1 Introduction

Training large language models (LLMs) often involves complex data pipelines. These pipelines handle large quantities of data, some of which might be sensitive. Recently, it has been shown that LLMs are susceptible to sentence-level membership inference attacks (Gu et al., 2023) and reconstruction attacks (Carlini et al., 2019), meaning that one may be able to infer which data was part of the training set, or in some cases, even reconstruct partial inputs by interrogating the model. As a result, this raises a prevalent problem of data removal from a trained LLM.

To this end, there has been growing interest in formalizing technical definitions of machine unlearning and designing machine unlearning techniques and evaluation metrics (Triantafillou et al., 2023, 2024). The goal of machine unlearning is to remove the influence of a subset of the original training data, the *forget set*, from a corresponding model. A naïve way to achieve it is to retrain the model from scratch on an updated training set (the *retain set*), that does not include the forget set. This approach is resource-intensive, and does not scale to the large models now in development.

Efficient alternatives in LLMs often rely on gradient ascent-based procedures, where one maximizes some loss on the data to be forgotten to reduce the influence of this data on the model predictions (Jang et al., 2022). However, there are a few issues that arise with this approach: (1) inherently, gradient ascent-based unlearning does not come with guarantees, and one needs a way to empirically evaluate the unlearning quality; (2) such unlearning methods do not only affect the forget set examples, but also come at a performance cost on the rest of the data.

Our work touches upon both of these issues. For the first issue, we propose two metrics for evaluating unlearning quality. The first metric, named *generalized exposure*, lower bounds unlearning quality under a particular unlearning definition (Triantafillou et al., 2023), but requires access to a

---

*School of Cybersecurity and Privacy, Georgia Institute of Technology. Work was started when the author was an intern at Google Brain, and affiliated with the National University of Singapore.
†Google DeepMind

*reference model*, that had never seen the forget set, to compute likelihoods. Another metric, *relative exposure*, is an approximation to the first, further estimating the likelihoods that would be computed by a reference model, only using the current model pre- and post-unlearning.

For the second issue, we present an extensive empirical evaluation, on LLMs, of how unlearning via gradient ascent differs for in- versus out-of-distribution examples. We visualize the trade-offs between unlearning quality as measured per our definitions, and performance on the rest of the data. We capture different patterns of these trade-offs depending on the difficulty of the examples in the forget set, and depending on the degree of memorization of these examples. Our contributions can be summarized as following:

- We propose a new metric for evaluating unlearning in generative models using a reference model that had never seen the unlearning data. Further, we propose an approximation to this metric that does not require having access to the reference model.

- Using our proposed metrics, we evaluate gradient ascent-based unlearning in large language models, and observe that unlearning out-of-distribution samples can be done nearly without affecting the BLEU score Papineni et al. (2002) just like in the reference model. In contrast, unlearning in-distribution samples affects the performance, unlike in the reference model. This indicates a weakness in gradient ascent-based unlearning, and suggests that simultaneous gradient descent on the retain data might be necessary.

- Finally, we evaluate whether measuring unlearning on a data point could be done using similar samples. We observe that similar examples in the training data are unlearned together with the ones on which unlearning is performed. Similar examples outside of the training dataset are almost not affected by this unlearning procedure. This explains why we observe performance degradation for in-distribution examples.

## 2 Preliminaries

Let $\Theta$ be a parameterized space of models (e.g., $\Theta = \mathbb{R}^d$ in the case of neural networks with $d$ parameters). For our purposes, we care only about the output distribution of learning algorithms. That is, if $Z^*$ denotes the set of finite sequences of input examples, and $\Delta(\Theta)$ denotes the space of distributions on $\Theta$, a learning algorithm will be viewed as a map $\mathcal{A} : Z^* \to \Delta(\Theta)$, and so running the algorithm on a size-$n$ dataset $S \in Z^n$ produces the model $\theta \sim \mathcal{A}(S)$.

In the context of autoregressive sequence models, such as LLMs, the set of training data $S \in Z^*$ consists of samples $x \in Z$, each a sequence of tokens, $x = (x_1, \ldots, x_k)$. A model $\theta$ defines conditional distributions on the next token $x_i$ given all previous tokens $x_{1:i-1}$, denoted $f(x_i|x_{1:i-1};\theta)$. For a fixed model $\theta$, we consider its *output* on a given sequence of tokens $x = (x_1, \ldots, x_k) \in Z$ to be $f(x;\theta) = \prod_{i=1}^k f(x_i|x_{1:i-1};\theta)$, the probability it assigns to that sequence (in other words, the likelihood of $x$ under the model $\theta$).

Let $\mathcal{L}(\theta, x) = -\log(f(x;\theta))$ denote the negative log likelihood (NLL) of $x$. The training objective of language models we consider is based on that loss, averaged over $x \in S$. It is minimized using gradient-based iterative algorithms. Although it is immaterial to this work, training can often be viewed as stochastic gradient descent (or some variant) applied to the objective $\theta \mapsto \mathbb{E}\,\mathcal{L}(\theta, X)$, where the expectation is taken over $X$ sampled from $S$.

### 2.1 Unlearning

Given that a learning algorithm has produced a model $\theta \sim \mathcal{A}(S)$, the goal of unlearning is to remove the influence of a subset $F \subseteq S$ of the training data. We call $F$ the *forget set*, $S \setminus F$ the *retain set*.

There are many ways one might formalize unlearning. We consider the following definition of unlearning (Sekhari et al., 2021; Gupta et al., 2021; Neel et al., 2021)[3]:

**Definition 2.1.** An algorithm $\mathcal{U}$ is a *worst-case $\varepsilon$-unlearner (for $\mathcal{A}$)* if, for every training set $S$, forget set $F \subset S$ of fixed size, and measurable subset $B \subseteq \Theta$, letting $\theta_{\mathcal{U}} \sim \mathcal{U}(\mathcal{A}(S), F)$ and

---

[3]Note that the cited papers usually have another parameter $\delta$ that accounts for a shift in Equation (1). In our case $\delta = 0$. Further, it is common to consider a type of a "publish" function that allows to compare a distribution over other quantities, potentially obtained by post-processing ("publishing") the output $\mathcal{A}(S)$.

$\theta_{-F} \sim \mathcal{A}(S \setminus F),$

$$e^{-\varepsilon} \Pr(\theta_{-F} \in B | S, F) \leq \Pr(\theta_{\mathcal{U}} \in B | S, F) \leq e^{\varepsilon} \Pr(\theta_{-F} \in B | S, F). \tag{1}$$

For any distribution over training data $S$ and forget sets $F$, we say $\mathcal{U}$ is an *on-average $\varepsilon$-unlearner* if Equation (1) holds when the probabilities are unconditional.

We refer to a sample $\theta_{-F} \sim \mathcal{A}(S \setminus F)$ as a *reference model*. Definitions of unlearning vary in a number of ways, including in terms of what information is available to the unlearning algorithm. The role of access to (statistics of) the training data is studied by Sekhari et al. (2021) .

In this work, we will study unlearning algorithms that operate by performing gradient ascent on the NLL loss, averaged over the forget set $F$.

## 3 Evaluation of Unlearning in Large Language Models

Fix a pair of algorithms $\mathcal{A}, \mathcal{U}$. Let $\mathcal{G}$ be the set of measurable functions from $\Theta$ taking values in $[0, 1]$. For training data $S$ and forget set $F \subseteq S$, the smallest $\varepsilon$ satisfying Equation (1) is

$$\varepsilon_{S,F} = \sup_{g \in \mathcal{G}} \left( | \log \mathbb{E}[g(\theta_{\mathcal{U}})] - \log \mathbb{E}[g(\theta_{-F})]| \right), \tag{2}$$

where $\theta_{\mathcal{U}} \sim \mathcal{U}(\mathcal{A}(S), F)$ and $\theta_{-F} \sim \mathcal{A}(S \setminus F)$. The supremum $\sup_{S,F} \varepsilon_{S,F}$ is the tightest parameter for the worst-case notion.

It follows that evaluating the argument in the r.h.s. of Equation (2) with any $g \in \mathcal{G}$ yields a lower bound on the unlearning parameter $\varepsilon$. Below we construct a function $g$ that we use to evaluate unlearning.

Let $F \subseteq S$ be a set of strings we want to forget. In addition, consider $n$ *reference strings* $R = \{r_i\}_{i=1}^n$, sampled from some given distribution, and that are not part of $S$ (or $F$). Recall that $\mathcal{L}(\theta, x) = -\log f(x; \theta)$ is the negative log-likelihood of a sequence $x$ under model $\theta$. Let

$$g(x; \theta, R) = \frac{1}{n} \sum_{j=1}^n \frac{\mathcal{L}(\theta, x)}{\mathcal{L}(\theta, x) + \mathcal{L}(\theta, r_j)}. \tag{3}$$

Each term $\mathcal{L}(\theta, x)/(\mathcal{L}(\theta, x) + \mathcal{L}(\theta, r_j))$ can be seen as a relaxation of the hard comparison $(\mathcal{L}(\theta, x) \leq \mathcal{L}(\theta, r_j))$ (or equivalently $(f(r_j; \theta) \leq f(x; \theta))$, as the NLL is monotonically decreasing). In aggregate, it represents the fraction of reference strings in $R$ that have an NLL higher than $x$. $g$ can be seen as a soft version (scaled to $[0, 1]$) of the rank of $f(x; \theta)$ among the probabilities of reference strings $\{f(r_j)\}_{j=1}^n$. A smaller value of $g$ indicates $x$ is more likely under $\theta$ (has a smaller loss) than elements of $R$, a larger value indicates it is less likely (has a larger loss). If $g(x; \theta, R) < \gamma$, then there are at most $2n\gamma$ elements $r_i$ of $R$ such that $f(r_i; \theta) > f(x; \theta)$ (and $\mathcal{L}(\theta, r_i) < \mathcal{L}(\theta, x)$). Similarly, if $g(x; \theta, R) > 1 - \gamma$, then at most $2n\gamma$ elements $r_i \in R$ satisfy $f(r_i; \theta) < f(x; \theta)$ (and $\mathcal{L}(\theta, r_i) > f(x; \theta)$).

We define *Generalized Exposure* of $x \in F$ relative to a set of reference strings $R$ to be

$$\text{GenEx}(x; \mathcal{A}, \mathcal{U}, F, S) = \log \mathbb{E}[g(x; \theta_{-F}, R)] - \log \mathbb{E}[g(x; \theta_{\mathcal{U}}, R)]. \tag{4}$$

Taking the absolute value of GenEx yields a lower bound on the worst-case epsilon in Equation (2) for a fixed $g$. One cannot compute the expectations in Equation (4) exactly, since the distributions of $\theta_{\mathcal{U}}$ and $\theta_{-F}$ are not tractable in a standard deep learning setup. In our experiments, we use a Monte Carlo estimate of the expectations in the generalized exposure metric to get an approximate lower bound on the unlearning quality. Such estimates are subject to variance. Alternatively, one could threshold the observed $g(x; \theta_{\mathcal{U}}, R)$, which would effectively correspond to choosing a different $g \in \mathcal{G}$ in Equation (2), and then use Clopper–Pearson confidence intervals for binomials to compute the confidence intervals of the estimates (Clopper & Pearson, 1934) (also see (Jagielski et al., 2020)).

**Exposure and memorization**[4]. Generalized exposure can be seen as an extension of the *exposure* metric that appeared in the memorization literature, introduced by Carlini et al. (2019). There, the

---

[4]We intend here a very restricted definition of "memorization": whether a generative model can be induced to generate near-facsimiles of some training examples when prompted with appropriate instructions. Models

authors inject secret *canaries* (i.e., strings generated randomly, from a different distribution than the regular data distribution) $C = \{c_i\}_i^m$ in the training set $S$. In our notation, $C = F$. In addition, $n$ reference strings $\{r_i\}_{i=1}^n$ are sampled from the same distribution. For each canary $c_i$, letting $\mathrm{rank}(l_i|\{l_j\}_j)$ denote the rank of $l_i$ in the set $\{l_j\}_j$, Carlini et al. (2019) define exposure as[5]:

$$\mathrm{Ex}(c_i; \theta) = \log_2(n) - \log_2(\mathrm{rank}(\mathcal{L}(\theta, c_i)|\{\mathcal{L}(\theta, r_j)\}_{j=1}^n)), \text{ or, equivalently} \tag{5}$$

$$\mathrm{Ex}(c_i; \theta) = -\log_2 \Pr_{j=1:n} [\mathcal{L}(\theta, r_j) \leq \mathcal{L}(\theta, c_i)]. \tag{6}$$

This metric is meant to capture how much the model memorized the canaries relative to the reference strings that were not seen during training.

Generalized Exposure uses a function $g$ (Equation (3)) that can be seen as a soft version of the comparison function used in the second formulation of exposure (Equation (6)). The reference strings it uses do not have to come from outside of the distribution of regular data in general, but we can consider $R = \{r_i\}_{i=1}^n$, as defined above, as a special case.

For a randomly generated string $r$ coming from the same distribution as $R$, and never seen during learning or unlearning ($\theta$ would be independent of them), $g(r; \theta, R)$ should be around ½, and each term in 4 of the form $-\log \mathbb{E} g(r; \theta, R) = \log(2)$. Similarly, the probability in Equation (6) will tend to ½, and exposure to $\log_2(2) = 1$.

When no memorization happens, Generalized Exposure for these randomly generated canary strings $C$ is zero. To see this, note that $\mathbb{E} g(c_i; \theta) = \frac{1}{2}$ under no memorization, and both sides of Generalized Exposure cancel out. For the exposure computation, the outcome of the comparison is $\frac{1}{2}$, giving $\mathrm{Ex}(c_i; \theta) = -\log_2 \frac{1}{2} = 1$. Under maximal memorization of $c_i$, the loss would be smaller than for all the reference strings, and thus $\mathrm{Ex}(c_i; \theta) \to \infty$. Similarly, the first term in the Generalized Exposure, $-\log \mathbb{E}[g(x; \theta_\mathcal{U}, R)]$ would tend to $\infty$ when $g(x; \theta_\mathcal{U}, R)$ gets arbitrarily close to 0 as $f(c_i; \theta_\mathcal{U})$ increases with more memorization relative to the reference strings.

**Membership inference attacks and differential privacy.** Jagielski (2023) connects the exposure metric from (Carlini et al., 2019) to differential privacy and so-called membership inference attacks. Recall that a training algorithm $\mathcal{A}$ is $\varepsilon$-differentially private (DP) if, for all $S$ and $S'$ that differ by one data point, and all measurable sets $B \subseteq \Theta$, $\Pr(\theta \in B) \leq e^\varepsilon \Pr(\theta' \in B)$, where $\theta \sim \mathcal{A}(S)$ and $\theta' \sim \mathcal{A}(S')$. One can interpret DP as a hypothesis test to assess whether the output of the algorithm was obtained by running $\mathcal{A}$ on $S$ versus $S'$. Kairouz et al. (2015) show that a particular computation based on false positive and false negative rates associated with this hypothesis test yields an estimate of $\varepsilon$ in the differential privacy definition.

Through this hypothesis test view, $\varepsilon$-DP can be connected to a version of so-called *membership inference attacks* (MIAs; see, e.g., Shokri et al. 2017), which attempt to identify whether a data point was or was not in the training set. Probably the most related MIA is a likelihood-ratio test (LiRA) introduced by Carlini et al. (2022a). LiRA is motivated by the connections to hypothesis testing, trying to determine whether the observed prediction is more likely to have been sampled from a model that was trained on the sample of interest or without. The authors choose to do a likelihood ratio test (motivated by the Neyman–Pearson lemma), assuming that the predictions for a given sample have a Gaussian distribution.

Inspired by the work by Kairouz et al. (2015) connecting differential privacy and MIAs, Triantafillou et al. (2023, 2024) propose to estimate $\varepsilon$ in the unlearning definition Equation (1) using false positive and false negative rates from a MIA perspective. In particular, letting $\{\pi\}$ denote all membership inference attacks, $\varepsilon$ in the unlearning definition above can be estimated as a supremum over $\{\pi\}$ of a function of upper and lower bounds of false positive/negative rates for $\pi$.

---

do not "contain" bit-wise or code-wise copies of their training data. Rather, if a model can be induced to generate very close copies of certain training examples by supplying appropriate instructions to guide the model's statistical generation processes then that model is said to have "memorized" those examples. This is an area of active ongoing research.

[5]They define it in terms of log-perplexity instead of NLL, but the only difference is a multiplicative $\log(2)$ factor, which is irrelevant in ranking and comparison.

## 3.1 Relative exposure

Generalized exposure requires computing the expected probability of $x$ under $\mathcal{A}(S \setminus F)$. Practically, having such a reference model may not be possible for computational and memory reasons. Here we introduce an alternative test that only requires access to the original model pre-unlearning.

As above, consider a set of reference strings $R$, and let $\theta_S \sim \mathcal{A}(S)$, $\theta_{-F} \sim \mathcal{A}(S \setminus F)$ and $\theta_{\mathcal{U}} \sim \mathcal{U}(\mathcal{A}(S), F)$. For each given $x$, we now randomly generate a second set of reference strings $R_x$, such that $\log \mathbb{E}[g(x; \theta_{-F}, R)] \approx \log \mathbb{E}\left[\hat{\mathbb{E}}_{r \in R_x}[g(r; \theta_S, R)]\right]$, where $\hat{\mathbb{E}}_{r \in R_x}$ denotes an empirical mean over the elements in $R_x$. In theory this is a complex task, and once again requires access to the reference model. In practice, however, we will simply choose $R_x$ so its elements are close to $x$ under some similarity metric (working in the embedding space), but not part of $F$; $R_x$ can contain examples from some auxiliary set (public data, held out data, etc.), that do not belong to the training set $S$. It is also possible to define a common $R_x$ for all $x \in F$. By choosing such a set, we ensure that $\theta_S$ does not depend on $R_x$, just like $\theta_{-F}$ does not depend on $x \in F$. Further, when the forget set is small and does not affect the predictions on $R_x$ through $\theta_S$ too much, we can expect our approximation to be more accurate.

Substituting this approximation to Equation (4), we get an alternative to Generalized Exposure that does not use $\theta_{-F}$. We define *Relative Exposure* of $x$ relative to $R, R_x$, as

$$\text{RelEx}(x; \mathcal{A}, \mathcal{U}, F, S, R, R_x) = \log_2 \mathbb{E}\left[\hat{\mathbb{E}}_{r \in R_x}[g(r; \theta_S, R)]\right] - \log_2 \mathbb{E}[g(x; \theta_{\mathcal{U}}, R)]. \qquad (7)$$

## 3.2 Memorization and example difficulty

When evaluating unlearning, we group examples in the forget set based on measures of the extent to which an example has been memorized and of the example's difficulty.

More carefully, let $\theta_S \sim \mathcal{A}(S)$ and $\theta_{-F} \sim \mathcal{A}(S \setminus F)$. For any fixed example $x \in F$, we define its *memorization* as

$$\log \mathbb{E}[f(x; \theta_S)] - \log \mathbb{E}[f(x; \theta_{-F})].$$

This is similar to the definition of memorization given by Feldman (2020) for classification tasks, which measures the difference between the probabilities of predicting the right class depending under $\theta_{-F}$ and $\theta_S$.

We define the *difficulty of any fixed example* $x \in F$ as $\mathbb{E}[-\log f(x; \theta_{-F})]$, i.e., expected perplexity under the reference model.

Note that all the expectations here are taken over the randomness of the training process (e.g., SGD noise, minibatch noise).

## 3.3 Unlearning by gradient ascent

Based on Equation (1), one could achieve *exact* ($\varepsilon = 0$) unlearning by retraining from scratch without the forget set $F$. This is not practical for large language models, due to resource constraints. Another common approach is to perform gradient ascent on the loss over $F$, or/and gradient descent on the loss over $S \setminus F$. Other alternatives have been proposed in the literature (Patil et al., 2023; Meng et al., 2022a,b), but a gradient ascent/descent-type procedure is still a common component in all of them.

While this approach is fairly efficient, and usually implemented with only a small number of gradient updates, it is not guaranteed that the obtained model after unlearning via gradient ascent/descent has truly forgotten the samples. Further, there is no set heuristic for the number of gradient updates to run during unlearning. This technique, thus, hinges on being able to assess how unlearned a set of examples is for a given language model.

## 3.4 Unlearning in- versus out-of-distribution samples

For a reference model, unlearning, which is equivalent to not training on, out-of-distribution samples should not affect the overall performance, meaning that the models $\theta_S$ and $\theta_{-F}$ should perform

similarly. When the forget set contains in-distribution samples, then the effect depends on the size of the forget set relative to the training set. We focus on the typical case where the size of the forget set is small enough, and both models, $\theta_S$ and $\theta_{-F}$, perform similarly under the BLEU score. Thus a good unlearning algorithm should be able to unlearn without any observable trade-offs between unlearning quality and overall performance, as measured by the BLEU score.

## 4   Experiments

We evaluate the trade-offs between unlearning quality and performance on LLMs. We demonstrate that unlearning more memorized or more difficult examples is more damaging for the overall model performance. We also examine how neighbouring examples are affected by unlearning. Finally, we show that our relative exposure metric captures unlearning quality as well as the generalized exposure metric, thus showing a way to assess unlearning without having a reference model.

### 4.1   Experimental setup

**Models and datasets.** We train a transformer model (T5-base with 220 million parameters (Roberts et al., 2022)) on "WMT14 En-De", a well-known language translation dataset that contains sentence pairs in German and English (Bojar et al., 2014). We train for $45,000$ training steps with batch size of $128$ on examples from the training split. We evaluate the task performance of the translation task using the BiLingual Evaluation Understudy (BLEU)) score (Papineni et al., 2002). Our models have a BLEU score of around 26, having a clear gist but with grammatical errors.

To avoid training numerous models, we only perform full training of two models:

- A *reference model*, of weights $\theta_{-F}$ is trained on the full standard training split $T$ of the dataset mentioned above, without any additional example from a forget set.

- A *subject model*, of weights $\theta_S$, is trained on a dataset $S$ made of $T$ and the concatenation of all the potential forget sets $F_{...}$ defined below.

We consider an unlearning method based on gradient ascent (Section 3.3). Following Jang et al. (2022), we use a batch size of 32 when unlearning a set of $512$ examples, giving us $16$ unlearning steps to go through for the entire forget set considered. During each unlearning experiment, we consider one single forget set $F$, and perform unlearning only on its examples. We always compare the resulting unlearned model with the same, shared reference model. Even though the reference model is only trained $T$, which is a subset of the retain set of any given experiment (the full retain set would include the forget sets for the other experiments), we consider it a suitable approximation, as the ignored examples form only a small fraction of the training set.

**Out-of-distribution forget sets generation.** We generate out-of-distribution (OOD) canaries by sampling alpha-numeric characters uniformly at random, forming a sequence of fixed length (10 characters). We create three disjoint sets $F_{...}^{\text{OOD}}$ of 512 OOD canaries each, as well as a set $R^{\text{OOD}}$ of 10,000 reference strings from the same distribution. To study the effect of the number of repetition on unlearning, these sets are incorporated in the training set $S$ with different frequencies: canaries in $F_{\times 1}^{\text{OOD}}$ are seen only once during training, the ones in $F_{\times 10}^{\text{OOD}}$ ten times, and $F_{\times 100}^{\text{OOD}}$ a hundred times.

**In-distribution forget sets generation.** We generate sets of in-distribution (InD) examples by randomly selecting examples from the validation split of the dataset, so that we can train the reference model on the full training split $T$, and these examples do not appear even once in its training set. We create three disjoint sets $F^{\text{InD}}$ of 512 in-distribution examples each, as well as a set $R^{\text{InD}}$ of $3,003$ reference strings formed from the test split. All of these sets of examples are disjoint. Similarly to the OOD canaries, these sets are incorporated in the training set with different frequencies. For the main subject model considered ($\theta_S$), $F_{\times 1}^{\text{InD}}$ is seen only once, $F_{\times 10}^{\text{InD}}$ ten times, and $F_{\times 100}^{\text{InD}}$ a hundred times. Appendix A.2 also considers a model trained on a different training set $S'$, where different validation examples are used in forget sets, see that section for details.

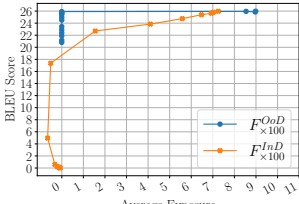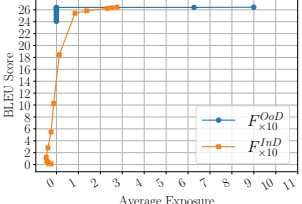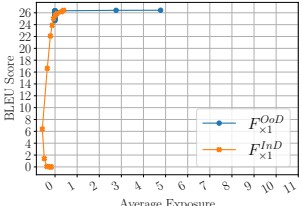

Figure 1: **In- vs. out-of-distribution (canary) trade-off.** Trade-off between the generalized exposure (Exposure) and the task performance (BLEU score) when unlearning the subject model ($\theta_S$) at 45,000 steps, with in-distribution sets and canary sets repeated 100 times (left), 10 times (middle), and 1 time (right) during training.

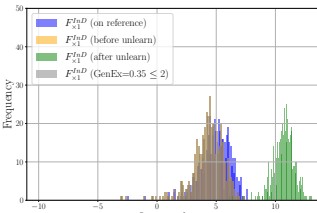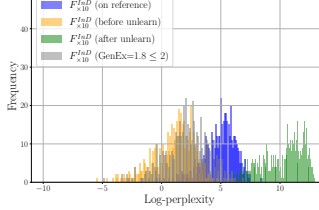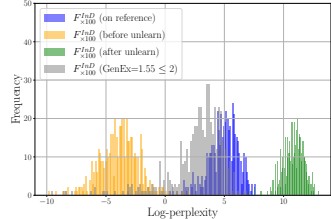

Figure 2: **Distributions of perplexities.** Perplexities of different sets of in-distribution examples under the subject model (before unlearning, post-unlearning and when exposure is low) and the reference model. Columns left to right: in-distribution example perplexities when the subject model was trained by repeating these examples 100 times (left), 10 times (middle), 1 time (right).

## 4.2 Memorization vs. performance trade-offs

Our evaluated method of unlearning modifies the model by performing gradient ascent, as a result it might degrade the model's accuracy on the test set. We first evaluate the trade-off between the effectiveness of unlearning under generalized exposure and the task performance on the unlearned model (Figure 1). At every unlearning step, we measure the *average* exposure of the canary, and, respectively, forget set. On these checkpoints, we compute the BLEU score on the test set.

Our first observation is that unlearning of canary data in one pass does not degrade the performance as much as unlearning in-distribution samples even when these are repeated as often. The average exposure value of the canaries also does not fall below 1 in one pass, meaning the canaries are still twice as less surprising to the model than other random samples unseen in training. The average exposure of the InD samples, however, falls to the minimum value. The reason is that unlearning InD examples affects the perplexities of other similar examples (Section 4.4), whereas for out-of-distribution, unlearning does not affect as much the other canaries' perplexities. This explains why the in-distribution examples have a much faster drop in exposure, as well as task performance.

**Different Frequencies.** In Figure 1, we observe that the more repeats of the in-distribution sample sets, the higher the (average) generalized exposure is before unlearning (top right point of each orange curve). A similar effect is visible for the exposure of OOD between the OOD $\times$ 1 and the OOD $\times$ 10 curves, although it is not visible in the OOD $\times$ 100 because the estimate of exposure is limited by $\log_2 |R^{\mathrm{OOD}}|$. In Appendix A.2, we also evaluate how a different number of repetition of the *same examples* of the in-distribution sets affect the trade-off. Despite the three randomly-selected InD sets having a different distribution of perplexities under the reference model (as shown in Figure 2), the qualitative results are not affected by which set is repeated a given number of times.

**Distribution of perplexities.** We check how the perplexities of the in-distribution samples are affected before and after unlearning with respect to the reference model. We observe that the perplexities of the in-distribution set is reduced, but now the perplexities are skewed, not resembling at all the distribution on the reference model (Figure 2). We also plot the distribution of perplexities when the exposure is below a certain threshold which results in distributions that are closer to

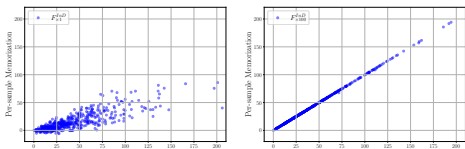 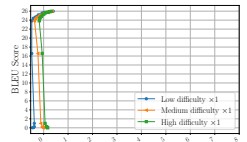 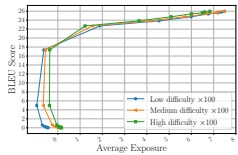

Figure 3: **Per-sample memorization vs. difficulty.** The memorization vs. difficulty for each sample in the forgets sets that repeat ×1, and ×100. Difficulty and memorization become correlated with number of repeats.

Figure 4: **Difficulty vs. trade-offs.** Measure the trade-offs of unlearning examples of low, medium and high difficulty. Harder examples have slightly better trade-offs.

the reference. This suggests an early-stopping strategy for unlearning could benefit in-distribution examples with more evidence of this effect at lower thresholds in Appendix A.4.

## 4.3 Per-sample difficulty vs. memorization

We empirically evaluate the relationship between the difficulty of in-distribution examples and memorization. In Figure 3, we plot the memorization and difficulty of each example in the forget sets. The per-sample difficulty has a weak correlation with the per-sample memorization when the InD set is repeated once, but the correlation becomes strong with the number of repeats. We also cluster the in-distribution examples into 3 sets of low, medium and high perplexity based on their difficulty (Figure 4), and find that harder examples have slightly better trade-offs (see details in Appendix A.3).

## 4.4 Unlearning effects on other points

We highlight that unlearning InD examples has an impact on other similar examples. We find similar examples by computing the $L_2$-distance in the embedding space of each point in the forget set on the reference model (see Appendix A.6). In Figure 5, we plot the memorization vs. performance trade-offs (as we unlearn $F_{\times 100}^{\text{InD}}$) for both the set $F_{\times 100}^{\text{InD}}$ and a set of similar examples from $F_{\times 1}^{\text{InD}}$. The average exposure of the similar set decreases, without having to do unlearning. This explains why unlearning damages the performance of the model, since the model may forget other examples. Despite this, the effect of unlearning on similar examples outside of the training set is not significant. Thus, unlearning may affect examples that are more memorized as opposed to just similar examples.

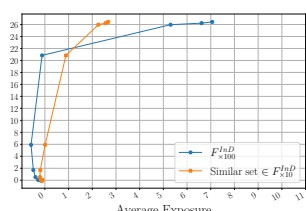

Figure 5: Unlearning affects the average exposure of similar examples.

## 5 Related Work

Recent work on unlearning in LLMs has focused on developing effective unlearning algorithms and robust evaluation metrics to assess the degree of unlearning achieved. We give a brief overview of the most relevant work here, and point interested readers to Appendix A.7 for more related work.

**Unlearning benchmarks and evaluation metrics.** Several works propose leveraging evaluation metrics of memorization with the aim to provide better unlearning methods in LLMs (Jang et al., 2022; Barbulescu & Triantafillou, 2024). Our work aims to work with a worst-case $\varepsilon$-unlearner (Definition 2.1) and can be seen as complementary to these approaches. Our experiments also point to stark differences between in- and out-of-distribution memorized data. An orthogonal unlearning approach is by removing of training data from the weights (Meng et al., 2022a; Patil et al., 2023).

**Memorization in LLMs.** Whereas our work targets memorized data unlearning, a range of other memorization notions and concerns have been studied in LLMs (Lehman et al., 2021; Ippolito et al., 2022; Carlini et al., 2021; Choquette-Choo et al., 2021; Lukas et al., 2023).

## 6 Conclusion

In this work, we propose a generalized exposure metric for evaluating unlearning. We find instances where gradient ascent-based techniques are insufficient for unlearning without destroying the model's performance. We explain this through the effect of unlearning on similar data.

**Acknowledgments**

We thank Daniel M. Roy and Eleni Triantafillou for feedback on various drafts of this work. This project used computational resources on Google Cloud Platform provided by Google, we would like to thank Danat Pomeranets in particular for support. Teodora was supported in part by the National Research Foundation Singapore under its NRF Fellowship Programme [NRF-NRFFAI1-2019-0004], by the Crystal Centre at National University of Singapore and its sponsors, and a Google PhD fellowship.

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

# A Appendix

## A.1 Experimental Setup Details

The training dataset is available in TensorFlow Datasets as "wmt_t2t_translate". It has three splits: a train split $T$ with $4,592,289$ examples, a validation split of $3,000$ examples, and a test split with $3,003$ examples. We use the validation split for selecting the InD sets and the test split for the reference strings for InD.

**Remark on out-of-distribution sets generation.** Note that our approach to generating canaries differs from that in (Carlini et al., 2019). There, the canaries are generated with a fixed string prefix (or template) such as "My secret is:" and a randomly-generated string suffix $c$, sampled from a randomness space $c \sim \mathcal{C}$, e.g., the alpha-numeric strings of length 10. These canaries aim to mimic accidental personal identifiable information (PII) in the training data, where the sensitive information was a unique string of characters, such as a social security number. However, having many canaries sharing the same template in the training set means that the model could learn to detect this pattern, and share some representation between canaries. This can be especially troublesome in the context of evaluating unlearning: decreasing the likelihood of a given canary could decrease the likelihood of the template, and that of the other canaries, leading to over-estimation of the effectiveness of an unlearning method.

## A.2 Variations among different in-distributions sets

To evaluate more directly how the number of repetitions of the *same examples* of the in-distribution sets affect the trade-offs, we train a second subject model, of weights $\theta_{S'}$, on a training set $S'$ containing:

- the training split $T$ of the dataset of interest;
- the same OOD forget sets as in $S$: $F_{\times 1}^{\mathrm{OOD}}$, $F_{\times 10}^{\mathrm{OOD}}$, and $F_{\times 100}^{\mathrm{OOD}}$;
- different in-distribution forget sets, made out of the same examples but with different frequencies: $F_{\times 1}'^{\mathrm{InD}}$ contains the same examples as $F_{\times 100}^{\mathrm{InD}}$ but repeated only once, $F_{\times 10}'^{\mathrm{InD}}$ contains the same examples as $F_{\times 1}^{\mathrm{InD}}$, and $F_{\times 100}'^{\mathrm{InD}}$ as $F_{\times 10}^{\mathrm{InD}}$.

Regardless of the identity of the repeated set, we observe that the more repeats of the in-distribution sample sets, the higher the (average) generalized exposure before unlearning, as shown in Figure 12). We also do not observe a significant difference in their average generalized exposures.

We did not investigate that effect on out-of-distribution canaries. Since they were sampled from a uniform distribution and should be interchangeable, we do not expect the identity of canary examples repeated the same number of times to influence the exposure.

Despite the average exposure being the same, the distribution of the InD set perplexities are different. We illustrate the perplexities of the three sets of in-distribution examples on a model that was trained without them in Figure 13. The three InD sets have similar mean log-perplexities on the reference model, with differences in the spread of the distribution of their log-perplexities. Specifically, the mean and variance for $F_{\times 1}^{\mathrm{InD}}$, $F_{\times 10}^{\mathrm{InD}}$, and $F_{\times 100}^{\mathrm{InD}}$ is $(41.95, 1068.65)$, $(42.30, 1061.54)$, and $(42.94, 1266.64)$, respectively.

We also plot the distribution of perplexities under the training set $S'$ in Figure 14 which shows the variation between the different InD sets.

## A.3 Difficulty and memorization results

Figure 22 shows the difficulty and exposure trade-offs for the different model trained on $S'$ where we vary the number of repetitions of the same examples. Each sub-figure shows the unlearning of one set: $F_{\times 1}^{\mathrm{InD}}$, $F_{\times 10}^{\mathrm{InD}}$, and, respectively $F_{\times 100}^{\mathrm{InD}}$. The conclusion is that harder examples with more repetitions have slightly better trade-offs as it does not lead to "over-unlearning" (skewing the exposure to negative values). Similarly, we show that variations among different in-distributions sets yields similar observations regardless of the identity of the in-distribution set (Figure 22).

We compute the memorization of each sample in the InD sets based on their likelihood on the subject model (trained on all forget sets $F_{...}^{\mathrm{InD}}$, before any unlearning) and their likelihood on the reference model. We average the likelihood of each sample over 3 reference models $\theta_{-F} \sim \mathcal{A}(S \setminus F)$, trained under three different seeds.

Figure 15 plots the per-example difficulty and memorization for the $\theta_S$ after unlearning on one of the forget sets. We use the definitions in Section 3.1 for difficulty and memorization. We find that there is a weak correlation between these two for InD examples that repeat once, but the correlation gets stronger as the number of times the samples repeats increases.

**Subject Models** (trained on the same OOD sets, but different InD sets)

$\theta_S$ **trained on** $(F^{\mathbf{InD}}_{\times 1}, F^{\mathbf{InD}}_{\times 10}, F^{\mathbf{InD}}_{\times 100})$

$\theta_{S'}$ **trained on** $(F'^{\mathbf{InD}}_{X \times 1}, F'^{\mathbf{InD}}_{Y \times 10}, F'^{\mathbf{InD}}_{Z \times 100})$

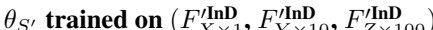

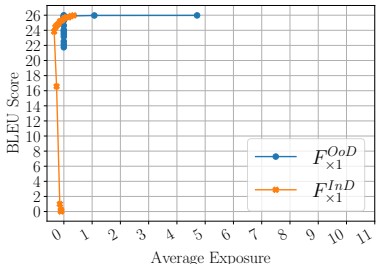

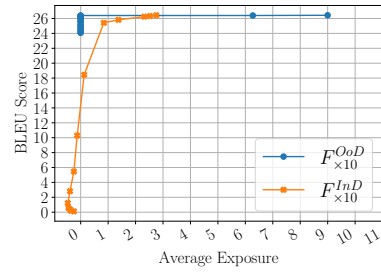

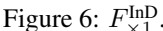

Figure 6: $F^{\mathrm{InD}}_{\times 1}$.

Figure 7: $F'^{\mathrm{InD}}_{X \times 1}$, same examples as $F^{\mathrm{InD}}_{\times 10}$.

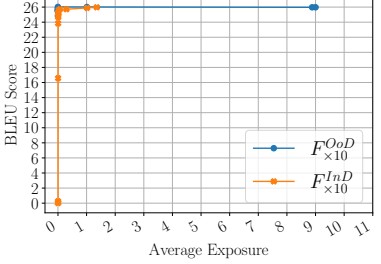

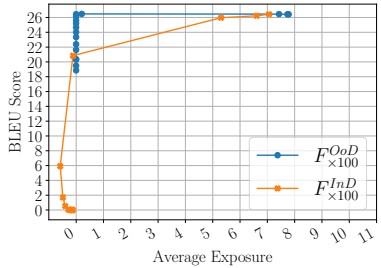

Figure 8: $F^{\mathrm{InD}}_{\times 10}$.

Figure 9: $F'^{\mathrm{InD}}_{\times 1}$, same examples as $F^{\mathrm{InD}}_{\times 10}$.

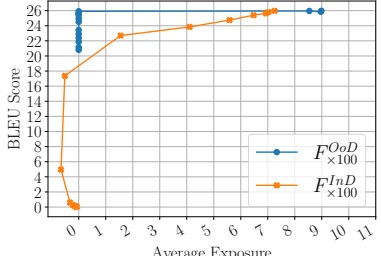

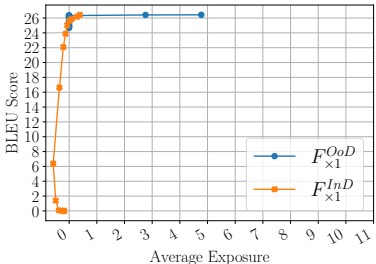

Figure 10: $F^{\mathrm{InD}}_{\times 100}$.

Figure 11: $F'^{\mathrm{InD}}_{\times 1}$, same examples as $F^{\mathrm{InD}}_{\times 100}$.

Figure 12: **In- vs. Out-of-distribution sets.** Trade-off between the generalized exposure (Exposure) and the task performance (BLEU score) when unlearning the subject models $(\theta_S, \theta'_S)$ at 45,000 steps, with OOD (canary) forget sets $(F^{\mathrm{OOD}}_{\times 1}, F^{\mathrm{OOD}}_{\times 10}, F^{\mathrm{OOD}}_{\times 100})$, and in-distribution forget sets $(F^{\mathrm{InD}}_{\times 1}, F^{\mathrm{InD}}_{\times 10}, F^{\mathrm{InD}}_{\times 100})$ for $\theta_S$ (resp. $(F'^{\mathrm{InD}}_{\times 1}, F'^{\mathrm{InD}}_{\times 10}, F'^{\mathrm{InD}}_{\times 100})$ for $\theta_{S'}$).

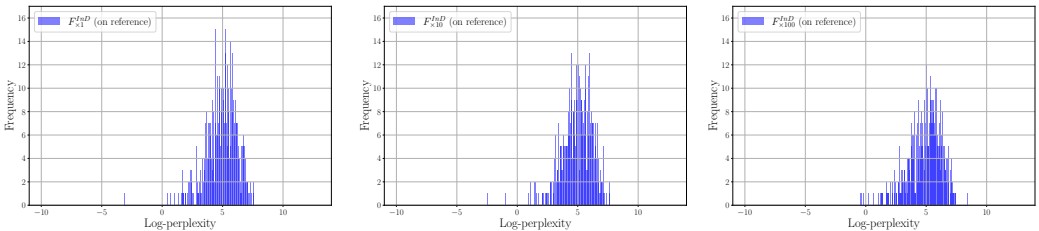

Figure 13: We show the distribution of the log-perplexities of the different sets of in-distribution examples used in our experiments. The perplexities were computed on the reference model that was trained on the language translation dataset (wmt-t2t, de-en), without OOD canaries or InD samples. We see clear differences, despite which the example frequency vs unlearning results were not different among these different groups.

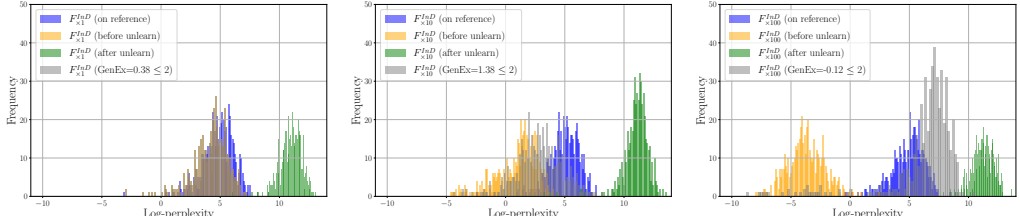

Figure 14: **Distributions of perplexities.** Perplexities of different sets of in-distribution examples under the subject model (before unlearning, post-unlearning, and when the exposure is before a threshold of 2), and the reference model when the training set is $S'$ (different frequencies for the same set of examples). Columns left to right: in-distribution example perplexities when the subject model was trained by repeating these examples 1 time (left), 10 times (middle), 100 times (right).

### A.4 Distribution of perplexities at low exposure

We note that the unlearning a number of steps may result in negative exposure, a sign of "over-unlearning" which skews the distribution of perplexities of the forget sets on the unlearned subject model compared to the distribution of perplexities of the forget sets on the reference model. We show what happens when we set the exposure threshold to $0.5$ in Figure 23. The extent of this effect depends on the number of repeats of the forget set.

### A.5 Relationship between relative exposure and generalized exposure

The relative exposure metric is a more computationally efficient one, since it does not require access to a reference model. We want to study whether it is a good proxy for the generalized exposure metric. For this, we take the subject model trained on $S$ and plot it together with the generalized exposure before unlearning (at the $45,000$ training step) and after unlearning. We observe that the relative exposure is a good proxy for the generalized exposure (Figure 24).

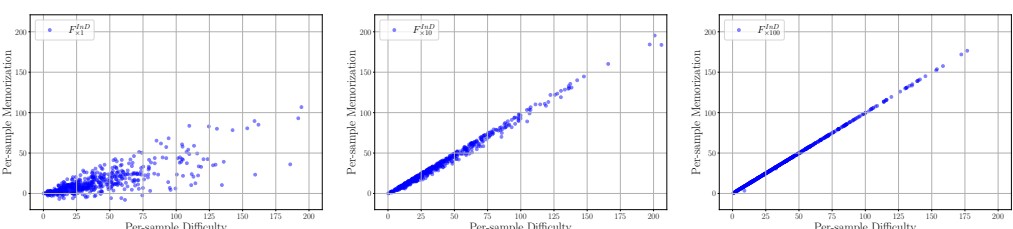

Figure 15: **Per-sample memorization vs. difficulty** The more times a forget set repeats, the more its difficulty is correlated with its memorization.

**Subject Models** (trained on the same OOD sets, but different InD sets)

$\theta_S$ **trained on** $(F^{\mathrm{InD}}_{\times 1}, F^{\mathrm{InD}}_{\times 10}, F^{\mathrm{InD}}_{\times 100})$   $\theta_{S'}$ **trained on** $(F'^{\mathrm{InD}}_{\times 1}, F'^{\mathrm{InD}}_{\times 10}, F'^{\mathrm{InD}}_{\times 100})$

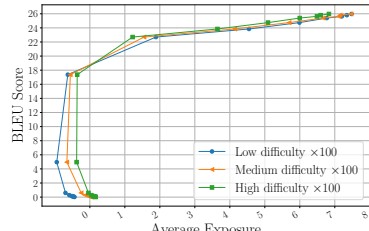 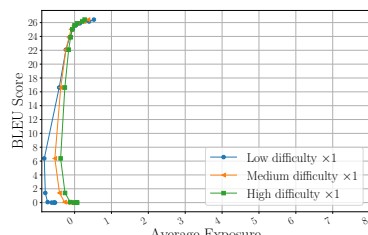

Figure 16: $F^{\mathrm{InD}}_{\times 100}$.   Figure 17: $F'^{\mathrm{InD}}_{X \times 1}$, same examples as $F^{\mathrm{InD}}_{\times 100}$.

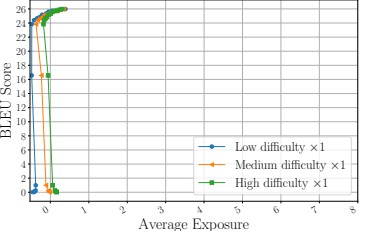 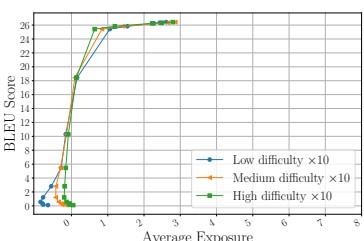

Figure 18: $F^{\mathrm{InD}}_{\times 1}$.   Figure 19: $F'^{\mathrm{InD}}_{\times 10}$, same examples as $F^{\mathrm{InD}}_{\times 1}$.

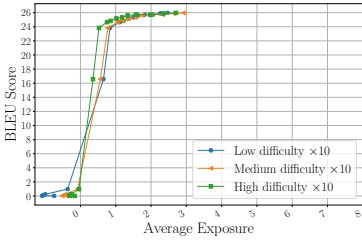 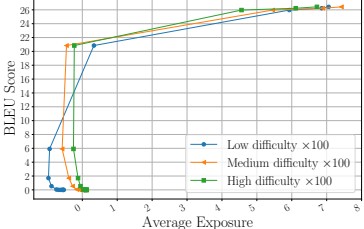

Figure 20: $F^{\mathrm{InD}}_{\times 10}$.   Figure 21: $F^{\mathrm{InD}}_{\times 100}$, same examples as $F^{\mathrm{InD}}_{\times 10}$.

Figure 22: **Difficulty vs. trade-offs.** For each set of in-distribution examples, we cluster them by difficulty using the perplexities on the *reference* model. Examples with a higher perplexity are considered harder. The trade-off is computed for each unlearning step on the main model. Harder examples have a better trade-off between the unlearning effectiveness and the performance of the unlearned model.

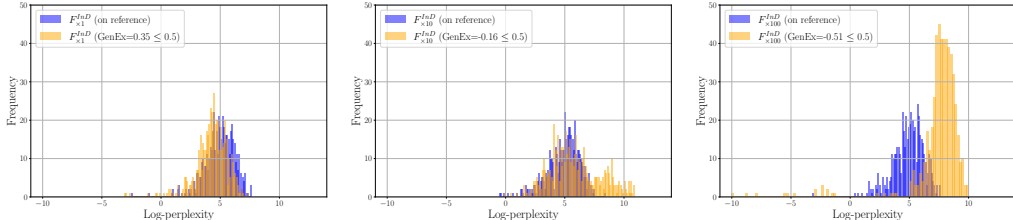

Figure 23: **Distributions of perplexities at low exposure.** Perplexities of different sets under the subject model at the first unlearning step which results in an average exposure lower than a threshold of $0.5$ (orange), and the perplexities under on the reference model (purple). In the top rightmost figure, we observe a phenomenon of "over-unlearning" (when the exposure becomes negative) which brings the two distributions of perplexities further apart.

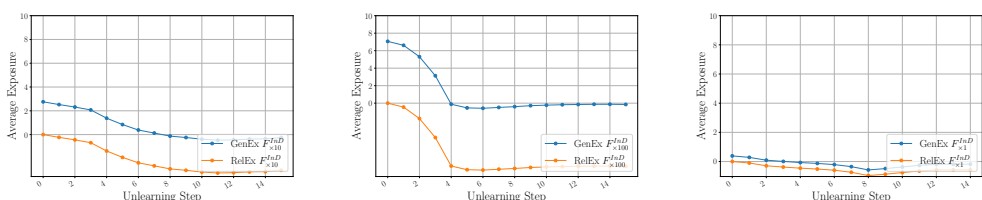

Figure 24: **Relative vs. generalized exposure** We find that relative exposure is a good proxy for the generalized exposure for in-distribution data.

## A.6 Effect of unlearning on similar points

To investigate this, we find similar examples (the top-10) from the set of examples that repeat ten times ($F_{\times 10}^{\text{InD}}$) to the forget sets $F_{\times 100}^{\text{InD}}$ and $F_{\times 1}^{\text{InD}}$. For simplicity, we compute the $L_2$-distance between the embeddings of each point in the forget sets on the reference model. Our similar sets consist of the union of the top-10 closest examples from $F_{\times 1}^{\text{InD}}$ for all examples $F_{\times 100}^{\text{InD}}$ and, respectively, $F_{\times 1}^{\text{InD}}$. Concretely, this resulted in $424$ examples for the set of examples that repeat once, and $421$ for the set of examples that repeat 100 times.

We then measure the average generalized exposure on the similar set as we unlearn the forget set $F_{\times 100}^{\text{InD}}$ and, respectively, $F_{\times 1}^{\text{InD}}$, for 16 training steps. We plot the tradeoffs between exposure and performance on the forget set and on the similar sets in Figure 25. We can see that similar examples are unlearned as well by performing unlearning on the forget set, even before unlearning impacts the model's utility.

We want to see how unlearning the forget set also influences examples outside of the training set, i.e., the reference set $R^{\text{InD}}$. We use the same methodology as above, and pick the top-10 closest examples from $R^{\text{InD}}$ to our two forget sets. This results in $571$ for $F_{\times 100}^{\text{InD}}$ and $591$ for $F_{\times 1}^{\text{InD}}$. To start with, the exposure of examples outside the training set is small. The effect of unlearning of the forget set on these examples' exposure is unnoticeable, though we do observe a small decrease of exposure (up to $0.1$ for the case shown in Figure 5). Similarly, we show that the effect of $F_{\times 1}^{\text{InD}}$ on the closest reference examples is very small. For the reference set and the forget set that is repeated only once, we observe the same phenomenon: the exposure of the reference set is not affected by the unlearning of the samples in the forget set (Figure 25). We also validate our main observation of the effect of unlearning on similar points in the training dataset on a different subject model, trained on a training dataset $S'$. The training has the same forget sets but with different number of repetitions.

## A.7 Related Work

**Unlearning benchmarks and evaluation metrics.** (Lynch et al., 2024) propose eight distinct evaluation metrics that go beyond standard loss measures on the forget/retain set, and try to capture internal model changes, as well as the impact on downstream tasks. The authors measure robustness to jailbreaks and finetuning, other extraction techniques, undesirable side effects, etc. Shi et al.

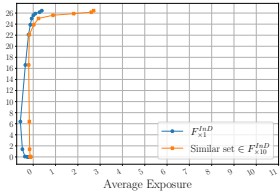 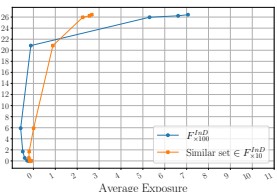 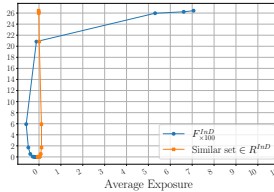

Figure 25: **In-distribution vs. similar in-distribution examples.** Trade-off between the generalized exposure (Exposure) and the task performance (BLEU score) when unlearning the subject model on set $F_{\times 1}^{\text{InD}}$ and $F_{\times 100}^{\text{InD}}$ (left to right). Unlearning in-distribution examples affects the exposure of other similar examples from the training dataset (left-most two plots), while not affecting the exposure of unseen examples from the reference set (right-most plot). We can see that the examples in the reference set are not affected by unlearning.

(2024) propose a new machine unlearning evaluation benchmark, MUSE, focusing on assessing 6 desired properties of unlearned models, such as verbatim memorization, scalability with forget sets size, etc. The TOFU benchmark paper (Maini et al., 2024) introduces a new task and dataset for evaluating specific training data unlearning in large language models. Jang et al. (2022) introduce an "extraction likelihood" metric for measuring unlearning quality in LLMs: they look at the average completion accuracy of a sequence of tokens when a varying length prefix was provided as a prompt. The authors also studied gradient ascent-based unlearning, and found that to be more effective when unlearning sequentially in batches rather than all at once. They also report differences in how easy it is to unlearn depending on the source of the forget set. While Jang et al. (2022) also points to differences in the effectiveness of unlearning between different forget datasets, they do not further explore how similar examples are affected by gradient ascent (as our work does). TOFU focuses on a "Task of Fictitious Unlearning" where models are trained on fictional author profiles and then must unlearn a subset of those profiles. The paper provides a dataset of these profiles, metrics to assess unlearning efficacy, and baseline results from existing unlearning algorithms. All of the work above aims to identify and assess desirable properties of unlearned models for general or specific tasks, but do not directly work with $\varepsilon-$unlearning definition as in Definition 2.1. The way to measure $\varepsilon-$unlearning as proposed in our work can be viewed as complementary to these other approaches.

Barbulescu & Triantafillou (2024)[6] leverage memorization information for unlearning by proposing an unlearning method that differentiates textual sequences based on their memorization level, as in (Jang et al., 2022). The memorization in this work is captured by tracking reconstruction of the exact tokens in a sequence, which is different from the definition used in our work. An unlearning algorithms is "successful" if memorization of a particular sequence of interest is reduced. Their work also introduces an MIA-like evaluation inspired by the neighborhood MIA concept.

**Memorization.** Several studies have explored different facets of memorization in LLMs, including verbatim memorization (Lehman et al., 2021; Ippolito et al., 2022), membership inference attacks (Shokri et al., 2017; Nasr et al., 2018; Salem et al., 2018; Choquette-Choo et al., 2021), exposure (Carlini et al., 2019), and extraction attacks (Carlini et al., 2021, 2022b). These works provide valuable insights into the extent and nature of information leakage in LLMs.

Hayes et al. (2024) highlighted the limitations of inexact unlearning evaluation methods like membership inference attacks. The authors show that current evaluation metrics for approximate unlearning can be misleading, creating a false sense of security. They call for more rigorous testing and a deeper understanding of how unlearning affects different data points..

**Removing information in Large Language Models.** Patil et al. (2023) consider information removal from the weights of a language model, which should protect against white box attacks. The authors focus on model editing techniques (Meng et al., 2022b,a), and show that even after editing the model to remove some sensitive information, they were still capable of extracting this information in a large fraction of cases. This paper also investigates how editing sensitive information affects the accuracy on neighbouring points using this information. They use the change of accuracy

---

[6]This work was carried out independently and concurrently with our work.

in the neighbourhood, a metric borrowed from (Meng et al., 2022b), to demonstrate that in many cases sensitive information was not properly removed.

**Memorization and membership inference attacks.**   Membership inference attacks (MIAs), first introduced for classification tasks, aim to evaluate to what extent a given datapoint can be traced back to be from a training set or not (Shokri et al., 2017). MIAs are now widely adopted in unlearning literature, as well as for studying memorization. Recently, membership inference attacks have been proposed for language models such as text classification tasks (Gu et al., 2023), (Mattern et al., 2023), and masked language models (Mireshghallah et al., 2022). The membership inference information can serve as a step towards extracting the training data. Carlini et al. (2019) showed that personal information can be extracted by generating numerous sentences from pre-trained language models and performing membership inference. Nakamura et al. (2020) considered an adversary with some prior knowledge of the patient that could employ a pre-trained masked BERT model to predict the masked personal information in the input clinical data. Lukas et al. (2023) showed that PII can be extracted from these models. Besides attacks, several mitigation strategies have been proposed for large language models such as ad-hoc practical defenses (Lee et al., 2021), as well as based on the rigorous framework of differential privacy (Ponomareva et al., 2022). Deferentially private training makes the model indistinguishable to an adversary (or user querying the model) up to one data record, or a fixed size group (group privacy). However, in unlearning, requests to delete samples may come for a batch of samples of varying size, perhaps even hundreds of these. As a result, differential privacy is not enough to support unlearning requests across all applications.

