# OpenReview forum: "Unlearning in- vs. out-of-distribution data in LLMs under gradient-based methods"
_NeurIPS.cc/2024/Workshop/SafeGenAi — SafeGenAi Poster_

### Official Review · Reviewer_wXjQ · 2024-10-08
**Review for Submission 148**

**Rating:** 7
**Confidence:** 3

**Review:**

The paper investigates the how unlearning for in- versus out distribution data effects generated LLMs, and propose new evaluation for unlearning. It shows that while unlearning out-of-distribution data will not affect models much, unlearning in-distribution data can hurts model's performance significantly. The experiment results support the hypothesis from several perspectives. Overall, I think it's a good work for the unlearning field.

To improve the paper for future submission, I encourage author to propose some novel algorithms to reduce the trade-off of unlearning for in-distribution data. Meanwhile, other evaluations for measuring the difference between unlearned model, reference models and subject models is not discussed. A good example will be the KL divergence between two models.

---

### Official Review · Reviewer_PwBa · 2024-10-09
**Review of "Unlearning In- vs. Out-of-Distribution Data in LLMs Under Gradient-Based Methods"**

**Rating:** 6
**Confidence:** 3

**Review:**

The paper *Unlearning In- vs. Out-of-Distribution Data in LLMs under Gradient-Based Methods* discussed the subject of "machine unlearning," a very important factor in data privacy and model retraining for large language models. This work is going to introduce a quantification method and evaluate the quality of unlearning, emphasizing how the unlearning of out-of-distribution data compares with the unlearning of in-distribution data. Their main focus lies in how they conduct the unlearning process through gradient-ascent-based methods, exploring the performance versus efficiency trade-off in these scenarios.

**Introduction and Motivation**

The work begins with a discussion of the need for machine unlearning in view of an emerging concern about privacy risks for models which have been trained on sensitive or proprietary data. While LLMs are increasingly important and find applications in decision-making tasks, membership inference and reconstruction attacks are a serious privacy concern, especially when there is personal or confidential data which could be accidentally learned by the model. The authors underline that traditional retraining methods cannot be scaled to the size and complexity of modern LLMs; thus, more efficient methods are needed.

The motivation is well-articulated and clearly shows the relevance of privacy issues within LLMs. This work will take advantage of works previously proposed, which explored the erasure of learned information with model performance preservation, in particular by leveraging techniques based on gradient ascent.

**Theoretical Framework and Methodology**

The core contribution of the paper is two metrics for the quality assessment of unlearning: **Generalized Exposure** and **Relative Exposure**. While Generalized Exposure requires a reference model, which has been trained without the data to be forgotten, the latter refers to a computationally efficient alternative that does not require such a reference model. These two measures quantify how much the model "remembers" or "forgets" data from the forget set and estimate the impact of unlearning on the model's performance.

Generalized exposure gives a measure over the model likelihood assignment to a sequence from the forget set compared to those that were not in the forget set. Relative exposure approximates this measure without needing a reference model, hence more practical for large scale applications.

It relies on unlearning by performing gradient ascent on the forget set, assessing its impact both on in and out-of-distribution data in a comparative manner. The novelty in this approach is not only in systematically comparing unlearning of various types of data but also in providing substantial empirical evidence regarding the efficacy and efficiency of the gradient-based methods.

 **Experimental Design**

This setup evaluates the unlearning of the T5-base model on the WMT14 English-German translation dataset. Both in-distribution and out-of-distribution examples are unlearned from the model. It is demonstrated through these experiments that the unlearning of examples belonging to OOD-the randomly generated canaries-result in no impact on the model's general performance, measured in BLEU scores.

The paper identifies the three key takeaways from these experiments and provides the following: 1. **OOD vs. InD Unlearning**: Unlearning OOD examples has more gradient ascent steps but has less effect on the performance of the overall model; on the other hand, InD examples lead to a fast degradation of performance.
2. **Impact on Similar Data**: Unlearning InD examples affects not only the forget set but also similar examples within the training data, hence the degradation in performance. This is less of an issue with OOD data.
3. **Difficulty and Memorization**: The difficulty and memorization of examples matter to unlearning. The examples which are harder or more memorized are harder to be unlearned and have a larger impact on the model's accuracy if unlearned.

**Results and Discussion**

These results indicate that the gradient ascent-based unlearning may be effectively used on OOD data without seriously affecting model utility. However, as far as InD data is concerned, unlearning becomes highly complicated. The degradation of performance while unlearning InD examples indicates some inherent challenges in the current methods of unlearning-especially those examples with high memorization.

They also point out that early stopping of the unlearning process at an optimal point may somewhat alleviate the performance trade-offs for InD data. However, the authors do note that the gradient ascent does not come with strong guarantees about completeness of unlearning and these methods need further improvements.
** Strengths**

1. **New Evaluation Metrics**: Generalized Exposure and Relative Exposure metrics introduce a concrete framework for evaluating unlearning quality in LLMs. These will likely prove helpful in future research on unlearning.

2. **Empirical Evidence**: The extensive empirical comparative study of InD and OOD unlearning fills a literature gap, hence supplying practical insights into the behavior of different types of data under unlearning.
3. **Scalability**: It is specially useful to consider scalable methods, such as gradient ascent-based unlearning, when the goal of machine unlearning is going to be applied in real applications where retraining from scratch will not be practical.

**Weaknesses**

1. **Skimpy Discussion of Ethical Implications**: While the paper discusses technical issues in great depth, the implications of unlearning in LLMs for society at large and its ethical considerations are not discussed in detail. Given the profound privacy risks indicated, further discussion on how these methods would affect data protection laws, ethical AI, and industrial practice in terms of compliance with the GDPR would have strengthened the work.
2. **Focus on a Single Dataset**: Experiments are limited to a single dataset, the WMT14 translation dataset. Though the derived insights are quite useful, extending experiments to other tasks, for example, classification and generative models, might allow generalizing the findings of unlearning methods more broadly.

3. **Inability to Guarantee Complete Unlearning**: The paper realizes that complete forgetting cannot be guaranteed with gradient ascent-based unlearning, and hence there are some limitations in the proposed approaches. Guaranteeing this appears to be an area that requires further research to provide more substantial guarantees.

** Conclusion**

The authors have, in this work, given a valuable contribution to machine unlearning by proposing new quality metrics for unlearning and performing empirical investigations of differences between unlearning InD and OOD. While the proposed gradient ascent-based methods are practical for OOD data, the challenges of unlearning InD data remain an open research question-especially those concerning avoiding performance degradation. The work is foundational in its approach to the evaluation of unlearning in LLMs; hence, further studies will be required to refine these methods and extend their applicability.

---

### Official Review · Reviewer_nZDh · 2024-10-09
**Need Comprehensive Context and Robust Experiments**

**Rating:** 4
**Confidence:** 3

**Review:**

This paper investigates machine unlearning techniques applied to large language models (LLMs). The study proposes a new metric, generalized exposure, to evaluate the quality of unlearning. While the paper addresses an important topic in AI, it could be improved in several areas.

Strength:
- The paper uniquely applies machine unlearning concepts to LLMs, exploring an important area of AI research.
- It provides a concise introduction to machine unlearning, helping set the research stage.
- The paper presents a well-formulated approach for unlearning in the context of LLMs.

Weaknesses
- The paper lacks a comprehensive background introduction, making it challenging for readers to fully understand the context and significance of the work.
- The experiments presented are not sufficiently robust to support the claims made in the paper.
- The paper does not adequately explain how unlearning contributes to the safety and security of LLMs.

First, The paper would benefit from a more thorough discussion of the security implications of machine unlearning in LLMs. For example, how does unlearning specifically contribute to safer LLM outputs? Can you provide concrete examples or scenarios where unlearning would be crucial for LLM safety?

Second, the experimental section needs significant improvement. How does the proposed method perform against sentence-level membership inference attacks and reconstruction attacks? Include comparative analyses with existing unlearning methods to highlight the proposed approach's advantages.

Third, to make the paper more accessible and easier to follow: (1) Provide intuitive explanations and examples of the model design. (2) Offer clear, real-world examples of how the proposed metric works.

This paper addresses a novel aspect of LLM research but requires more comprehensive context, robust experiments, and a clearer discussion of security implications to significantly advance the field.